# Age-Related Lysosomal Dysfunctions

**DOI:** 10.3390/cells11121977

**Published:** 2022-06-20

**Authors:** Lena Guerrero-Navarro, Pidder Jansen-Dürr, Maria Cavinato

**Affiliations:** 1Institute for Biomedical Aging Research, Universität Innsbruck, 6020 Innsbruck, Austria; lena.guerrero-navarro@uibk.ac.at (L.G.-N.); pidder.jansen-duerr@uibk.ac.at (P.J.-D.); 2Center for Molecular Biosciences Innsbruck, Innrain 58, 6020 Innsbruck, Austria

**Keywords:** lysosome, aging, senescence

## Abstract

Organismal aging is normally accompanied by an increase in the number of senescent cells, growth-arrested metabolic active cells that affect normal tissue function. These cells present a series of characteristics that have been studied over the last few decades. The damage in cellular organelles disbalances the cellular homeostatic processes, altering the behavior of these cells. Lysosomal dysfunction is emerging as an important factor that could regulate the production of inflammatory molecules, metabolic cellular state, or mitochondrial function.

## 1. Aging and Senescence

Aging is a process associated with the detriment of normal physiological functions, which leads to the manifestation of diverse diseases such as cardiovascular and neurodegenerative diseases, joint degenerative diseases, and metabolic diseases such as diabetes, among others [1].

The aging process has been associated in many tissues to an increase in the number of senescent cells, which is considered as a hallmark of organismal aging. This rise in the ratio of senescent cells contributes to the pathogenesis of age-related diseases [1].

Senescence is a physiological process involved in the suppression of tumors, which also participates in the development of organisms and in wound healing, where these cells transiently appear and are removed [2]. However, the permanence of these metabolic active cells in aged tissues promotes the dysfunctional remodeling of the tissue and foments inflammation, contributing to tissue decline [2].

Precisely, due to their active metabolism, senescent cells affect their environment and neighboring cells, through the secretion of matrix metalloproteinases, growth factors, chemokines, cytokines, and other immunomodulatory molecules. The composition of this set of factors, commonly denominated as senescence-associated secretory phenotype (SASP), changes depending on the type of cell and the senescence-inducing stimuli [3].

Cells can undergo senescence in response to various intrinsic and extrinsic stimuli, giving rise to different subtypes of senescent cells. Replicative senescence occurs when the intrinsic stimuli that induces senescence is telomere shortening. When senescence is induced by an external harmful stimulus such as exposition to oxidant molecules or ionizing irradiation, it is referred to as stress-induced premature senescence (SIPS) [4]. Oncogene induced senescence occurs when a prooncogenic mutation or a loss-of function mutation in a tumor-suppressor gene induces a strong antiproliferative response. Recently, other types of senescence have been identified, for example, mitochondrial dysfunction-associated senescence (MiDAS), in which mitochondrial dysfunction induces a type of senescence with a characteristic SASP that lacks the IL-1 inflammatory arm [5].

As evidenced, senescence is heterogeneous, and some markers present in senescent cells also exist in other physiological conditions. For this reason, the identification of senescence requires the detection of several factors. These factors include SASP molecules, cell cycle arrest markers, chromatin remodeling, and SA-β-gal activity, among others [3].

One of the main characteristics of senescent cells is their inability to proliferate since their cell cycle can be arrested by several pathways. The principal pathways that are studied in this context are p53/21 and pRb/p16. p53, considered the guardian of the genome, is stabilized by post-translational modifications in response to cell damage, particularly damage to the genome, favoring the expression of CDK inhibitors such as p21 and p16. This leads to the dephosphorylation and consequent activation of pRb, which represses the activity of the E2F transcription factor family, stopping the cell cycle progression. The accumulation of p21, and particularly, of p16, are essential for the maintenance of senescence [2,6].

Age-related dysregulations in the organization of chromatin occur in senescent cells, with different structural changes occurring in different genes. Heterochromatin areas are present in target regions of E2F, reinforcing the cell cycle arrest, while other parts of the genome such as pericentromeric regions display a loss of heterochromatin, associated with genome reorganization and aberrant transcription [3,7]. Additionally, an increase in histone y-H2AX foci associated with unrepaired DNA double-strand damage is a common characteristic of several types of senescent cells [2].

However, not only genomic damage has been reported as a feature of senescent cells. For instance, the nuclear membrane of different strains of human and murine cells is also impaired, with Laminb1 downregulation a common feature in senescent cells [8].

Additionally, senescent cells commonly present increased activity of the enzyme senescence-associated β-galactosidase detected at a pH of 6.0 (SA-β-gal). In fact, SA-β-gal is present in lysosomes, and in young cells, is detected at pH 4 and in senescent cells at a pH of 6 [9]. The fact that this enzyme works at a higher pH in senescent cells has not aroused much interest until recently, when damage to lysosomes associated with senescence and alterations in lysosomal acidification were evidenced [10,11,12].

Senescence is usually accompanied by other characteristics that can be linked to lysosomal function such as the loss of proteostasis, deregulated nutrient-sensing, mitochondrial dysfunction, and altered intercellular communication. Below, these hallmarks will be explained in the context of lysosomal dysfunction during aging.

## 2. Lysosomes

Lysosomes are heterogeneous organelles enclosed by a lipid bilayer and filled with hydrolytic enzymes. The lysosomes are traditionally described as the subcellular structures where the degradation of other organelles and macromolecules takes place, a fundamental process for maintaining cellular proteostasis [13].

There are several degradation processes in which the lysosomes are involved. If the substrate reaching the lysosomes comes from the extracellular environment, the degradation process is called endocytosis. If the material to be digested comes from the cell itself, the process is classified as autophagy. The lysosomes are also involved in plasma membrane repair through a mechanism called lysosomal exocytosis. Furthermore, degradation products from lysosomal catabolism can be sensed by metabolic complexes at the external side of its membrane, serving as a scaffold for metabolic regulation [13].

### 2.1. Lysosomal Structure and Components

To understand the lysosomal participation in these processes, the composition and structure of the organelle must be understood. The lysosome is delimited by a bilipidic membrane that encloses an acid lumen, where the degradative reactions take place. This lumen contains approximately 60 hydrolases that are active at acidic pH and participate in multistep catabolic processes to orchestrate the degradation of organelles and macromolecules to monomers [13]. Lysosomes have a role in protein degradation, since they contain proteases such as cathepsins. However, they are also involved in the degradation of other macromolecules such as lipids, which are catabolized by lipases such as the lysosomal acid lipase LIPA [14]. Nucleases such as RNase T2 or DNase II participate in nucleic acid degradation [15], and enzymes such as the acid alpha-glucosidase are important for polysaccharide degradation [16]. Macromolecules catabolized into monomers inside the lysosomes can be reused in anaplerotic reactions, connecting lysosomal function with cellular metabolism.

The lysosomal membrane is covered in its internal part with a large glycocalyx, a thick layer of polysaccharides that prevent lysosomal membrane self-digestion, avoiding acid content leakage into the cytosol [13]. The lysosomal membrane contains key proteins for the fusion of the lysosome with autophagosomes, endosomes, or with the plasma membrane as well as pumps and channels that are fundamental for the transport of ions and molecules and to maintain lysosomal acidification [17].

Lysosomal acidification is tightly controlled. These organelles are acidified mainly by the action of the v-ATPase, which hydrolyses ATP to pump protons into the lysosomal lumen. This process generates a transmembrane voltage that is dissipated by the action of ion channels [17]. The correct functioning of these ion channels is important not just to dissipate the electrogenic gradient, but also for v-ATPase to continue introducing protons against the gradient. Indeed, an increase in lysosomal pH can have detrimental consequences on the digestion carried out by lysosomal hydrolases, since these enzymes are active at low pH (pH 4–5).

The compensation of the electrogenic gradient is achieved either through the action of a cation leaving the lysosome or by an anion entering the lysosome. In fact, the entry of chlorine (Cl^−^) through channels such as the CLC-7 channel compensates for the entry of H+ in the acidification process, while cation channels such as TPRMLs and TPCs facilitate the efflux of K^+^, Na^+^, and Ca^2+^ [17].

Lysosomes are important calcium stores and the release of these ions is important to regulate endosome–lysosome fusion, autophagy, or lysosomal biogenesis [18,19,20]. Typically, lysosomal calcium efflux is regulated by Nicotinic acid adenine dinucleotide phosphate (NAADP), and the local calcium content allows the amplification of the signal increasing ER calcium release [21]. Moreover, contacts between the endolysosome and the ER have been described, being relevant for endosome maturation [18,22].

### 2.2. Lysosomal Biogenetic Pathways and Metabolic Integration

Lysosomal biogenesis is important for lysosomal adaptation to different situations and to replace disrupted or dysfunctional lysosomes. Similarly, under nutrient deprivation conditions, when autophagy is induced, lysosomal biogenesis is activated since the number of lysosomes must increase. Additionally, when a cell replicates, lysosomes have to be produced to be dispersed between the daughter cells. Diverse mechanisms participating in lysosome biogenesis have been described [23].

The mammalian target of rapamycin (mTORC) is a protein kinase involved in the regulation of cell growth and metabolism. mTORC is the main component of mTORC1 and mTORC2, two protein complexes that are distinguished by their accessory proteins. Aside from mTORC, mTORC1 contains the regulatory-associated protein of mTOR (RAPTOR), and mTORC2 is characterized by the presence of the rapamycin-insensitive subunit companion of mTOR (RICTOR) [24].

One mechanism that regulates lysosomal biogenesis depends on mTORC1 (mammalian target of rapamycin complex 1) activity, which converges in the activation or inhibition of TFEB (Figure 1). mTORC1 can sense the metabolic state of the cell. In nutrient-rich conditions, active mTORC1 binds with TFEB, the master transcriptional factor in the regulation of lysosome biogenesis and function, at the lysosomal surface [25]. mTORC1 phosphorylates TFEB, causing its binding, sequestration, and consequent inactivation by the regulatory 14-3-3 proteins in the cytosol. When the cell requires nutrients, mTORC1 is inactivated and TFEB translocates to the nucleus where it promotes the transcription of its target genes, thus enhancing lysosome biogenesis and autophagy [13,23]. This set of genes, involved in lysosome biogenesis, is known as the CLEAR (Coordinated Lysosomal Expression and Regulation) gene network. TFEB is the main regulator of this network, but other transcription factors from the MiT/TFE family to which TFEB belongs such as TFE3, MITF, and TFEC can also regulate the transcription of these genes [26]. TFEB also regulates the expression of v-ATPase subunits, so mTORC1 indirectly regulates v-ATPase activity and the acidification process [27]. The v-ATPase subunit association is also a tightly regulated process. Although this mechanism has been better studied in yeast, it is known that in mammals, the assembly of v-ATPase is promoted by amino acid starvation and by glucose starvation [28]. Furthermore, TFEB enhances the expression of genes related to lipid catabolism as a response to cell starvation [29]. TFEB also enhances the transcription of PGC1-α, which is the master regulator of mitochondrial biogenesis and participates in mitochondrial clearance [30].

mTORC1 can regulate autophagy through TFEB activity, which induces the expression of genes involved in autophagy. In contrast, when nutrients are available, active mTORC1 phosphorylates and inhibits the activity of the complex ULK-ATG13-EIP200, required for autophagosome biogenesis, inhibiting autophagy [31].

Another described mechanism that regulates lysosomal biogenesis depends on protein kinase C (PKC). Similar to what has been described for mTORC1, the activation of PKC leads to phosphorylation and the consequent inactivation of the GSK3β kinase, which, in turn, fails to phosphorylate TFEB [32]. Unphosphorylated active TFEB translocates to the nucleus, where it performs its function. Additionally, the activation of PKC promotes lysosomal biogenesis through a second mechanism, in which the activation of JNK2/p38 leads to the phosphorylation of ZKSCAN3, a DNA-binding protein that upon phosphorylation leaves the nucleus, where it was repressing the expression of lysosomal genes [32].

Precisely, PKC can be activated by mTORC2, which emphasizes the role of mTOR in lysosomal biogenesis. Although mTORC1 function is better understood than the one of mTORC2, some mTORC2 effectors have been identified. Thus, mTORC2 can phosphorylate AGC kinases [33] including protein kinase B (AKT) and PKC. mTORC2 responds to growth factors, with the PI3K pathway important for mTORC2 activation [33], which can phosphorylate PKC, influencing lysosomal biogenesis. More research is needed to elucidate how mTORC2 regulates lysosomal biogenesis in response to growth factors, however, it appears that lysosomal positioning regulates the activation of mTORC1 and mTORC2 [34].

Lysosomal biogenesis can also be actively repressed. MYC, a transcriptional factor involved in cell proliferation, can, under certain conditions, occupy the promoter regions recognized by TFEB, disturbing the transcription of lysosomal biogenesis-related genes. Additionally, MYC interacts with histone deacetylases in these same promoter regions, interfering with the epigenetic regulation of lysosomal biogenesis [23,35].

Lysosomal biogenesis, which depends on TFEB, is closely related to the regulation of autophagy, and for this reason, to the physiological state of the lysosomes. Therefore, dysregulation of lysosomal biogenesis influences the degradative mechanisms that occur inside the lysosomes. The main publications that study the lysosomes in the context of aging are focused on autophagy, since the lysosomal state is tightly interconnected with the autophagic flux [36,37,38].

However, lysosomes also participate in the regulation of other molecular pathways, themselves being a subject of study in pathophysiological conditions. For instance, incorrect function of lysosomes can lead to metabolic alterations given that lysosomes act as a scaffold for metabolic integration, especially affecting mTORC1 function, which, in addition to autophagy, influences downstream processes such as translation, lipid synthesis, or energy metabolism [24]. Similarly, lysosomal disruption has important consequences in other processes with lysosomal participation such as endocytosis, mitophagy, or lysosomal exocytosis [13].

## 3. Processes in which the Lysosome Participates

### 3.1. Endocytosis

Endocytosis is a cellular process in which cellular membrane engulfing allows for the internalization of external material. There are several types of endocytosis: phagocytosis, macropinocytosis, clathrin-mediated endocytosis, caveolin-mediated endocytosis, and clathrin-and caveolin-independent endocytosis [39]. In each type of endocytosis, different membrane receptors participate in the engulfing process. After plasma membrane engulfment, an endosome is formed. The newly formed endosome matures to an early endosome that mediates receptor recycling to the membrane. The transition to late endosome is coordinated by small GTPases following a maturation pathway. This maturation allows for the progressive acidification of the endosome, which finally fuses with the lysosome, where it reaches the most acidic pH [40].

A general downregulation of endocytosis during aging or senescence has been observed, and some components important for endocytosis regulation such as βPIX or GIT also seem to be downregulated in senescent cells. βPIX is a p21-activated kinase that seems to control membrane ruffling. In fact, the knockdown of βPIX induces senescence in human dermal fibroblasts. In this case, the senescence phenotype is accompanied by a suppression of clathrin-mediated endocytosis, linked to the cleavage of amphiphysin 1, an endocytic adaptor important for actin polymerization. This dysfunctional endocytosis seems to be linked with persistent activated integrin signaling, which can be important for the senescent phenotype, as integrin signaling inhibition prevents senescence [41]. There is no information on the lysosomal dysfunction repercussions in endocytosis during senescence [42].

### 3.2. Autophagy

Autophagy is a cellular catabolic process that is lysosomal-dependent, in which the cell degrades its own material. It can be classified into three types: microautophagy, chaperone-mediated autophagy, and macroautophagy [43]. Macroautophagy is characterized by the formation of double-membrane vesicles, called autophagosomes, which sequester the substrates to be degraded. It is the most studied process, and it is the only way of eliminating whole organelles, so it is crucial for the removal of damaged organelles.

Macroautophagy requires the formation of a phagophore as a starting point, which leads to autophagosome development. The phagophore does not form from the budding of a pre-existing membrane, but the nucleation point starts at the cytosol and the ER seems to participate in lipid donation, establishing a structure called omegasome [44]. Autophagosome biogenesis occurs in three steps: nucleation, expansion, and closure [43]. LC3 is a protein involved in substrate selection and its lipidation is an important process during autophagy initiation. LC3 associates with the lipid phosphatidylethanol-amine (PE) in the phagosome, where it is processed by Atg3 and Atg7. The phagophore, which is a double membrane structure, is elongated in a process that involves these ATG proteins in a procedure regulated by kinases such as ULK and PI3KC3-CI. Finally, the fusion with the lysosome depends on RabGTPases and SNARE proteins [43,45,46].

Several studies have found that autophagy declines with aging in different organisms such as Drosophila and mice [47,48]. Particularly in humans, the expression of several autophagic proteins such as Atg5, Atg7, or BECN1 show a decline with aging [49]. In fact, in many animal models, a premature aging phenotype is observed when the activity of genes related to autophagy is silenced either by knockout or knockdown [50]. Additionally, in mice, the knockout of some autophagy-related genes such as Atg5, Atg9, or Atg13 results in a non-viable phenotype [51]. In contrast, the promotion or restoration of autophagy is often accompanied with a lifespan extension [52]. In fact, defects in autophagy are related to the development of age-related diseases such as atherosclerosis [53].

When it comes to cellular senescence, the role of autophagy becomes controversial. Autophagy can modulate the homeostasis of pro- and anti-senescence factors, either facilitating or impeding the development of the senescent phenotype depending on the stimuli, its duration, or the cell type [37]. In oncogene-induced senescence, for example, the inhibition of autophagy seems to ameliorate senescence [54]. However, the inhibition of autophagy in normal proliferating cells can facilitate the production of ROS promoting senescence. The selective induction of autophagy and the elimination of GATA4, a transcription factor that regulates senescence, may be beneficial for senescence reduction [37].

### 3.3. Mitophagy and Mitochondrial Dysfunction

Lysosomes and mitochondria are tightly interdependent [55]. Several mechanisms have been proposed to explain how lysosomal function interferes with mitochondrial function during aging. Mitophagy has emerged as the main link between these organelles.

Mitochondria are double-membrane-bound organelles that play a major role in cellular metabolism since aerobic respiration takes place in these organelles. The TCA cycle and fatty acid oxidation generate reducing agents that donate electrons to the mitochondrial electron transport chain for ATP production. Aside from their function in bioenergetics, mitochondria are organelles that influence cell fate since many enzymes that regulate apoptosis such as BCL-2 proteins or cytochrome c are found in these cellular compartments [56].

When the mitochondria are damaged and the mitochondrial membrane potential decreases, the respiratory chain does not work efficiently and ROS are produced. ROS generate oxidized proteins and lipids that further exacerbate mitochondrial dysfunction. These impaired mitochondria have to be removed, and the main mechanism to eliminate whole organelles is macroautophagy. The selective degradation of deficient mitochondria through macroautophagy is called mitophagy [56]. Thus, lysosomes are directly involved in the maintenance of mechanisms of mitochondria quality control.

Many authors have studied the effects of mitochondrial impairment on the lysosomes [57,58,59]. In T cells, the impairment of mitochondrial respiration enhances lysosomal biogenesis through TFEB. However, these lysosomes seem disrupted, as evidenced by lysosomal alkalinization, cathepsin B activity reduction, and lysosomal sphingolipid accumulation. Moreover, this same study demonstrated that mitochondrial impairment is linked with pro-inflammatory phenotypes [57]. This shows the important role played by lysosomal function in the induction of SASP on senescent cells, which are known for their partially dysfunctional mitochondria.

Moreover, the deletion of mitochondrial proteins such as AIF, OPA1, or PINK1, or the chemical inhibition of the electron transport chain result in lysosomal impairment. Antioxidant treatment can partially rescue the impaired lysosomes, suggesting that mitochondrial defects that favor ROS production lead to lysosomal defects [58]. Other authors suggest that, depending on the mitochondrial damage duration, lysosome biogenesis can be enhanced or downregulated. In particular, mitochondrial respiration dysfunctions lead to lysosomal biogenesis induction in short-term ETC inhibition. However, long-term respiration inhibition by rotenone, a complex I inhibitor, results in lysosomal biogenesis repression [59].

Similarly, in yeast, it has been found that lysosome-like vacuoles increase their pH during replicative aging, and this change affects normal mitochondrial function. A screening of relevant differentially expressed genes during this aging process identified VMA1 (a v-ATPase subunit) and VPH2 (an enzyme that participates in the v-ATPase assembly) as genes whose overexpression delayed mitochondrial impairment during aging. The overexpression of these genes has a pro-longevity effect, which is surprisingly not related to an enhanced autophagic flux but to the storage of neutral amino acids in the vacuole [60]. This shows the role of v-ATPase in the correct acidification and repercussion in the lysosomal function, which at the same time can resonate in the mitochondrial function.

Other authors have explored the repercussion of lysosomal inactivation on the mitochondria. Pyruvate seems to protect cells against senescence, as was already shown in the MiDAS [5]. However, in this study, the authors did not address how pyruvate could affect lysosomal function. Pyruvate deprivation promotes senescence and seems to enhance lysosomal inactivation through the acetylation of the v-ATPase, which leads to an accumulation of abnormal mitochondria. In this case, mitochondrial impairment is linked to mitophagy defects [61].

In yeast, it has been observed that the kinase Sch9 controls vacuolar v-ATPase. The lack of Sch9 is correlated with a more acidic cytosolic pH, which seems to be dependent on target of rapamycin 1 (TORC1), the mTORC1 homologue in yeast [62]. Genetic disruption of vma, a v-ATPase subunit, leads to mitochondrial impairment accompanied by iron–sulfur cluster deficiency. Increased iron uptake suppresses the mitochondrial dysfunction [56]. Iron is essential for mitochondrial function as it is used for the synthesis of cofactors that participate in oxidation–reduction reactions [63]. Thus, lysosomal function seems to be important for iron homeostasis.

Moreover, some lysosomal storage diseases that are the products of mutations in genes that code for specific lysosomal proteins are characterized by mitochondrial dysfunction phenotypes. For example, cells with mutations in lysosomal enzymes such as NPC1 display aberrant mitochondrial lipid content that influences metabolic processes such as glycolysis [64]. This shows that the correct catabolism in the lysosomes affects the function of other organelles such as the mitochondria, although these events do not always correlate with mitophagy, but mitochondrial metabolism may be affected.

### 3.4. Lysosomal Exocytosis

Lysosomes can fuse with the plasma membrane and secrete lysosomal content through exocytosis. This process was first believed to be reserved for specialized cells such as macrophages or melanocytes [65,66], but it has been observed that it occurs in all cells as a repair mechanism of the plasma membrane [67].

Most lysosomes are localized in the perinuclear area, but when the plasma membrane is damaged, lysosomes migrate to the cell periphery to fuse with the plasma membrane promoting the recovery of the membrane. The lysosomal fusion with the membrane is regulated by lysosomal calcium efflux through TRPML1. Precisely, lysosomal exocytosis is frequently studied by the presence of lysosomal proteins such as LAMP1 or TRPML1 in the plasma membrane [68].

Lysosomal exocytosis is a process that also allows to get rid of unprocessed materials and it has been described in the case of lysosomal enzymes. Cathepsin D has been found in extracellular media upon lysosomal alkalinization [69]. Other groups have reported that lysosomes that fuse with the plasma membrane can get rid of autophagosomes [70]. LC3 lipidation is important for lysosomal exocytosis, so its regulation must be closely linked to autophagy [70].

## 4. Lysosomal Age-Related Dysfunctions

As evidenced, lysosomal function is very important for cellular homeostasis, not only because of the recycling functions it exerts on damaged molecules and dysfunctional organelles, but also because of the impact it has on metabolic pathways or on organelles such as the mitochondria. Most studies are not focused on lysosomal function, but instead cover fields in which the lysosome is involved such as autophagy or mTORC1 status. However, some studies have reported dysfunctions associated with the lysosome in the context of aging (Figure 2).

Precisely, mutations in enzymes related to lysosomal function such as in hydrolases or in lysosome membrane channels lead to lysosomal storage diseases, which are usually characterized by the increased accumulation of protein aggregates, leading to symptoms that resemble neurodegenerative diseases [71,72].

### 4.1. v-ATPAse Dysfunction and Lysosomal Alkalinization

Although some alterations have been described in luminal enzymes, the lysosomal membrane also has its relevance when it comes to aging. The correct v-ATPase function and lumen acidification has been correlated with delayed aging in yeast [60,62]. Many mutations in the v-ATPase subunits have been associated with neurodegenerative disorders, and animal models carrying mutations in v-ATPase subunits show lysosomal acidification problems accompanied by accelerated aging [73]. Cells lacking v-ATPase subunits also display mitochondrial dysfunctions, highlighting the important relation between lysosomes and mitochondria [74,75].

For instance, in *C. elegans*, it was recently demonstrated that several lysosomal genes such as v-ATPase subunits or lysosomal hydrolases are downregulated during aging. Moreover, these genes were found to be upregulated in long-lived mutants [76]. Furthermore, in *C. elegans*, the increase in lysosomal pH in older worms correlates with diminished proteostasis and the ceasing of reproduction. DAF-16, a gene previously known to regulate longevity, induces the expression of v-ATPase subunits and lysosomal acidification [77], evidencing the importance of correct lysosomal acidification for the promotion of longevity. Additionally, *C. elegans* lysosomes can release molecules that modulate longevity. In particular, the overexpression of the lysosomal acid lipase LIPA-4 promotes longevity, inducing the nuclear translocation of the fatty acid binding protein LBP-8 [78], indicating that lysosomal metabolism is crucial for lysosomes to act as signaling organelles, influencing the lifespan.

Precisely, the lysosomal state can modulate the metabolism, not only in *C. elegans*, but also on human cells. Recently, it has been described that the disruption of the lysosomal membrane during cellular senescence causes lysosomal alkalinization and, importantly, cytosolic acidification, which ultimately modifies cellular metabolism to counteract this acidification. In particular, acidic cytosolic pH is counteracted by increasing glutaminolysis [79]. Thus, lysosomal alkalinization results in metabolic cell adaptation, which foments the survival of senescent cells, denoting the important role lysosomes play in cellular metabolism and cell fate.

In macrophages, lysosomal exocytosis, which is an important functional mechanism, is propitiated by lysosomal alkalinization [68]. It would be interesting to know whether lysosomal exocytosis is also affected when lysosomal disruption and alkalinization occurs on senescent cells.

Importantly, the inhibition of lysosomal acidification by bafilomycin promotes iron deficiency, which results in impaired mitochondrial function and increased inflammation, linking correct lysosomal acidification to the functional mitochondrial state. Iron supplementation rescues these effects in cultured neurons and in a mouse model of impaired lysosomal acidification induced by the knockout of acid α-glucosidase, an essential enzyme for glycogen catabolism [75]. This evidences that lysosomal dysfunction led to imbalances in the compartmentalization of metabolites and ions, which have repercussions on other cellular functions.

### 4.2. Lysosomal Amino Acid Storage and Ion Homeostasis

Lysosomal dysfunctions lead to the accumulation of metabolites, and lysosomal storage diseases are an example of how the incorrect function of one lysosomal enzyme leads to storage problems, which conclude in signs and symptoms at the systemic level [80].

In yeast, aging has been linked to the incorrect storage of neutral amino acids in the vacuole [60]. Moreover, it has been proposed that the disruption of amino acid compartmentalization into the lysosome-like vacuole during yeast aging, especially impairment of vacuolar cysteine storage, causes cysteine accumulation in the cytosol. Cytosolic cysteine limits the iron bioavailability and leads to mitochondrial respiration impairment while cysteine depletion restores mitochondrial function [81].

Lysosomes receive iron through the endocytic pathway. The lysosomal enzyme STEAP3, whose activity depends on correct lysosomal acidification, is essential for iron reduction into Fe^2+^, the form that is incorporated in iron-containing proteins. Moreover, lysosomes participate in the turnover of ferritin, the protein that stores iron inside the cell, and of mitochondria, organelles that contain great quantities of iron. Therefore, iron homeostasis is tightly regulated by the lysosome, and iron disturbances promote the accumulation of ROS as Fe^2+^ can react with hydrogen peroxide, inducing the formation of highly reactive species through the Fenton reaction [82]. Moreover, iron is indispensable for processes such as oxygen transport or collagen biosynthesis, and it is part of many mitochondrial complexes, as iron–sulfur clusters are essential for oxidation–reduction reactions [63,83]. For this reason, the dysregulation of iron homeostasis contributes to cardiovascular [84] and neurodegenerative diseases [85,86,87], among others.

Since disturbances in iron homeostasis lead to ROS formation, it is not surprising that iron imbalance contributes to the aging process. In fact, old people suffer from problems of absorbing iron at the systemic level, while an increase in iron at the cellular level has been observed in this same population. Indeed, senescent cells have a 10-fold increase in iron compared to young cells [88]. This iron accumulation during senescence is associated with impaired ferritin degradation in the lysosome and increased expression of cell cycle inhibitors [89]. Furthermore, lysosome involvement in iron homeostasis is linked to a type of cell death called ferroptosis. Ferroptosis, which has been linked to aging [90], is promoted by ROS generation, associated with iron disturbances in the lysosome.

Other ions have been associated with lysosomal function during aging. For example, decreased potassium levels in yeast are consistent with increased acidity in the vacuole and lifespan extension [91]. Ion storage depends on the correct lysosomal function and acidification, but further research is needed to elucidate how lysosomal storage affects the aging process.

### 4.3. Lipofuscin

Lipofuscin, hydrophobic yellow-brown granules composed of oxidized lipids and proteins that accumulate in the lysosomes, is a common feature of senescent cells [3]. ROS generation has been linked to lipofuscin accumulation, as this granulated material tends to incorporate metals such as iron, contributing to oxidative reactions [92]. ROS damages proteins that are prone to unfold, leading to protein aggregation, thus contributing to the generation of adducts of oxidized cellular molecules. The insolubilization and crosslinked structure of lipofuscin impedes its correct degradation, and lipofuscin structures tend to accumulate in lysosomes. However, upon the inhibition of macroautophagy, lipofuscin also accumulates in the cytosol [93]. It has been proposed that lipofuscin remains attached to the proteasome, which is unable to degrade these cross-linked aggregates [94].

During aging, autophagy and subsequent lysosomal degradation of lipofuscin is impaired, which promotes further aggregation of these granules. Additionally, not only do impaired lysosomes contribute to lipofuscin accumulation during senescence, but the lack of cell division impedes the distribution of these aggregates between the daughter cells, as would occur in proliferative cells. Moreover, the increase in ROS production and lipofuscin accumulation promotes mitochondrial dysfunction, which further exacerbates lysosomal impairment in a positive feedback mechanism [95]. Additionally, lipofuscin accumulation enhances the activity of caspase-3 and promotes lysosomal membrane disruption, having been linked to NLRP3 inflammasome activation and necroptosis induction [92,96].

### 4.4. Inflammation and Cell Death

Senescent cells show increased mitochondrial mass, and these mitochondria are often dysfunctional. Typically, the mitochondria of senescent cells display low membrane potential, lower mitochondrial ATP production, and increased ROS production [56]. The decreased levels of mitophagy observed in many senescent cells supports a possible mechanism that explains the increase in dysfunctional mitochondria. Lysosomal dysfunction impedes the correct degradation of mitochondria, exacerbating mitochondrial ROS production which, in turn, increases lysosomal damage in a feedback fashion. Both mitochondria and lysosomes have been correlated to SASP production. In fact, mtDNA depletion seems to induce MiDAS with a characteristic SASP pattern that lacks the IL-1 inflammatory arm [5], while lysosomal disruption seems to induce IL-1 activation, enhancing NLRP3 inflammasome function [97].

NLRP3 is an important protein complex for the innate immune system. The two-signal model is proposed to explain NLRP3 activation [98]. The first or priming signal normally consists of PAMPs (pathogen-associated molecular patterns) or DAMPs (damage-associated molecular patterns), which lead to NF-κB pathway activation. The second or activating signal can be supplied by diverse inputs such as extracellular ATP, changes in ionic flux, mitochondrial dysfunction, or lysosomal disruption, among others [98].

Lysosomal disruption with Leu-Leu-Ome triggers NLRP3 activation [99] and it has been hypothesized that the disruption of these organelles and the subsequent cathepsin B release into the cytosol can activate NLRP3 inflammasome, and this process is important for pro-IL-1β processing [97], linking the loss of lysosomal integrity to inflammation. Indeed, some authors have shown that treating senescent cells with lysosomal function inhibitors such as leupeptin induces inflammation [57].

Lysosomal membrane permeabilization is not only involved in the activation of pro-inflammatory pathways, but it has been suggested to be involved in the mechanisms of cell death, and recently, it has also been associated with pathophysiological conditions [12]. Indeed, some authors have proposed that lysosomal membrane permeabilization in association with lysosomal quality control mechanisms can determine cell fate, since lysosomal components that are released into the cytosol can trigger the activation of diverse cellular pathways [100]. Several studies have suggested that lysosomal damage is related to apoptosis, necroptosis, and ferroptosis. Apoptosis, the main type of programmed cell death, which is characterized by cytochrome c release from mitochondria, can be fostered by the release of lysosomal cathepsins, which can degrade Bcl-2 [101,102], one of the proteins involved in mitochondrial membrane stability during apoptosis. Similarly, cathepsin D release has been associated to the activation of RIPK1 during necroptosis [103,104], a type of cell death though to be non-programmed, but now known to be regulated by receptor-interacting protein kinases (RIPK). Ferroptosis is a regulated cell death type in which intracellular iron incites the formation of ROS. Lysosomes are closely related to ferroptosis, since these organelles constitute one of the main iron storage places. Lysosomal membrane disruption seems to foster the activation of this cell death pathway [105].

Lysosomal damage seems to be critical for the development of different cell death pathways, and cell death is an important mechanism for the development and maintenance of age-related diseases such as neurodegenerative or cardiovascular disorders [106,107], making lysosomal damage an interesting field to research in this context.

## 5. mTORC and Senescence

mTORC1 seems to be activated during aging and in some senescent cells [108,109]. This activation is evidenced by the phosphorylation of mTORC1 downstream targets. It has been reported that in replicative senescent fibroblasts, p70S6K undergoes phosphorylation by the activity of mTORC1 [110]. Moreover, it has been found that constitutive mTORC1 activation induces premature senescence in fibroblasts carrying tuberous sclerosis complex (TCS) mutations [111]. S6 kinases, phosphorylated and activated by mTORC1, are observed in aged muscle [112] and in the brains of Alzheimer’s disease patients [113].

mTORC1 seems to be fundamental for the processing and activation of some SASP factors such as IL-6, IL-8, or IL-1A, which are, in turn, fundamental players in the inflammation induction that accompanies aging (Figure 2). Rapamycin treatment, which inhibits mTORC1, diminishes the pro-inflammatory phenotype of senescent cells through a reduction in the IL-1A and IL-6 levels [114]. Senescence induction seems to increase mTORC1 activity, which could be related to SASP induction through the TOR-Autophagy Spatial Coupling Compartment (TASCC). It has been suggested that mTORC1 localizes at this TASCC compartment in the Trans-Golgi and favors the synthesis of IL-6 and IL-8 cytokines [115]. Precisely, lysosomes originate from the Trans-Golgi network and mTORC1 activation occurs on the lysosomal membrane. Further studies are necessary to elucidate the relevance of the role of the mTORC1-lysosome on inflammation induction, but several authors have pointed out the role of lysosomes on inflammasome activation [80,96].

The mTOR pathway can positively or negatively regulate p53, being cell type and stress dependent [116]. In fact, mTORC1 and mTORC2 were reported to participate in the stabilization of cell cycle inhibitors (Figure 3). In MEFS, mTORC1 activation promotes the association of p53 mRNA with ribosomes, leading to p53 translation [117]. The overexpression of the microRNA miR-107 increases MTORC1 activity through PTEN inhibition, resulting in the activation of p16 [118]. Precisely, the loss of PTEN enhances p53 translation, triggered by mTORC1 [119]. Moreover, it has been reported that in PTEN-depleted cells, mTORC1 and mTORC2 bind and phosphorylate p53 at Ser15 [120]. This exhibits the participation of mTORC1 in senescent state regulation.

Precisely, S6K1, an mTORC1 downstream effector, has been related to MDM2 inhibition, avoiding p53 degradation, and subsequently promoting p53 stabilization, an important process for senescence induction [121]. S6K1 has also been reported to upregulate p16 [111] and to phosphorylate RICTOR, inhibiting mTORC2 and AKT [122]. Inhibited AKT is unable to phosphorylate and activate MDM2, promoting p53 stabilization [123].

In some cancer cells, mTORC1 activation through 4E-BPI phosphorylation stabilizes p21 [124]. In U2OS cells supplemented with branched-chain amino acids, p21 is also increased via mTORC1 activation, promoting cellular senescence in the presence of DNA damage-inducing compounds [125]. Similarly, it has also been shown that treatment with the mTORC1 inhibitor rapamycin decreases p53 translation, downregulating p21 and preventing AKT-induced senescence in human fibroblasts [126].

## 6. Lysosomal Opportunities for Intervention in Aging

Strategies that increase autophagic activity have been proposed as a mechanism to eliminate senescent cells. In particular, the suppression of mTORC1 extends the lifespan of many animal models, and inhibitors of this pathway such as rapamycin or torin1 have been developed [127]. Other molecules such as metformin have been shown to increase autophagy and ameliorate inflammation [128].

It is suggested that the treatment of senescent cells with mTORC1 inhibitors ameliorates the senescence phenotype in many cases [129,130]. It has been pointed out that although senescent cells undergo a permanent loss of proliferative potential because their cell cycle is arrested, their cellular growth pathways remain active. For instance, among the several pathways deregulated during senescence, the presence of ROS has been linked to cell cycle block, along with an active mTORC1 [109]. Rapamycin, a mTORC1 inhibitor, could prevent the permanent loss of proliferation when cells are arrested by p21 or p16 function. The proliferation of cells arrested by p21 or p16 upregulation in the presence of rapamycin resumed the proliferation capacity once rapamycin was removed, suggesting that mTORC1 activation is important for the achievement of senescence [129].

Importantly, mTORC1 inhibition prevents geroconversion, defined as the transition from quiescence to senescence, supporting the importance of mTORC1 activity to achieve senescence [130]. During cellular starvation, contact inhibition or hypoxia, cells do not undergo senescence, and it is hypothesized that mTORC1 inhibition prevents the conversion into a senescent state. Precisely, the stimulation of mTORC1 during contact inhibition favors the geroconversion [130].

Therefore, mTORC1 has become one of the principal targets for the development of senolytic compounds. Rapamycin blocks the activation of mTORC1, having been shown not only to have effects on senescent cells, but also in several organisms, where it is able to extend the lifespan. Mice fed with a diet containing rapamycin exhibits lifespan extension [131]. Additionally, genetic interventions targeting TOR influence the animal lifespan. For instance, in *C. elegans*, TOR RNAi silencing promotes a lifespan extension [132]. Similarly, p70S6K deletion in mice increases their lifespan [133].

Part of the beneficial effects of mTORC1 inhibition is thought to be due to an increase in autophagic flux. However, although rapamycin can directly influence lysosomal status, it has been reported that the lysosomal calcium channel MCOLN1 is also directly activated by rapamycin [134]. This channel is important for lysosomal biogenesis since Ca^2+^ efflux through this channel creates a Ca^2+^ microdomain, which activates the phosphatase calcineurin, which, in turn, dephosphorylates TFEB and allows for its transport to the nucleus [19]. MCOLN1 status during senescence is unknown, but loss-of-function mutations of this gene lead to a lysosomal storage disease called Mucolipidosis IV [135]. Further research is needed in the context of aging and senescence, but it seems that TFEB activation ameliorates age-related diseases [136,137].

Since TFEB was reported as the master regulator of lysosome biogenesis, many efforts have been made to identify activators that promote lysosomal function and autophagy, since the correct function of lysosomes is the key to promoting longevity. Many compounds able to activate and upregulate TFEB affect TFEB downstream of the inhibition of mTORC1, but one compound called C1 has been identified as a direct activator of TFEB. Accordingly, C1 binds TFEB and promotes its nuclear translocation [138].

In some neurodegenerative disease models, increased TFEB function through genetic intervention induces the activation of autophagy and a re-establishment of proteostasis, ameliorating the protein accumulation characteristic of these diseases. In Huntington’s disease, studies with in vitro models have shown that TFEB activation reduces HTT protein aggregation and decreases disease-related symptoms in mice [136]. In Parkinson’s disease, TFEB activation also ameliorates α-synuclein toxicity through the stimulation of autophagy [137]. Alzheimer’s disease is also related to impairment of proteostasis, and an accumulation of autophagosomes in the brains of Alzheimer’s patients has been observed. TFEB activation could also have beneficial effects in this disease, since a reduction in tau pathology and neurodegeneration in a mouse model has been observed upon increased TFEB activity [11,139].

In contrast, TFEB overexpression seems to promote the development of some tumors [140,141,142]. Considering this, it is possible that the therapeutic use of TFEB activation may be limited by its oncogenic potential.

Other strategies do not focus on remodeling senescent metabolism, but on killing senescent cells. This is the case of senolytics, drugs that aspire to specifically remove senescent cells by targeting a mechanism that is normally upregulated in a specific senescent phenotype. However, as in the case of cancer treatments, it is difficult to address a treatment to a specific type of cell and avoid off-target effects. In recent years, an elegant proposal has been made, where the increased SA-β gal activity from the senescent lysosome was used to activate a senolytic compound [143]. The senolytic prodrug enters the cell and is sequestered in the lysosome, where SA-β gal can catalyze the cleavage of the prodrug, therefore selectively eliminating senescent cells.

## 7. Conclusions

The correct degradation of macromolecules is important to integrate distinct metabolic routes. The correct function of hydrolases is key to degrading polymeric macromolecules (e.g., proteins) to their monomeric building blocks (e.g., amino acids), which can be sensed by specialized proteins associated with mTORC1. In addition, these monomers act as substrates for anabolism, and their compartmentalization is important to regulate cellular metabolism, and thus, other cellular functions. Specific studies to investigate the role of lysosomal disruption in intracellular storage problems are needed. In some senescence models, lysosomal dysfunction and/or disruption has been reported, however, it is not known to which extent this event directly affects the phenotype of these cells. Similarly, lysosomal membrane disruption causes lysosome alkalinization and cytosolic acidification, which could also alter normal cellular functions besides glutaminolysis.

The emerging function of these organelles on the metabolic regulation or molecule storage must be studied in more depth, especially the relation between lysosomes and mitochondria. As has been observed, mitochondrial dysfunction can lead to lysosomal problems, and simultaneously, lysosomal dysfunction can disembogue in problems associated with mitochondria. Because of this, the understanding of signaling mechanisms existing between these organelles, especially the ones associated with monomer storage, is of special interest, since it can be fundamental to understand how senescence develops. However, future studies must discriminate between distinct mechanisms that take place in the lysosome, and in this way, it will be determined whether the relationship between lysosomes and mitochondria goes beyond mitophagy and can be associated with storage changes, lysosomal permeability, or other mechanisms.

## Figures and Tables

**Figure 1 cells-11-01977-f001:**
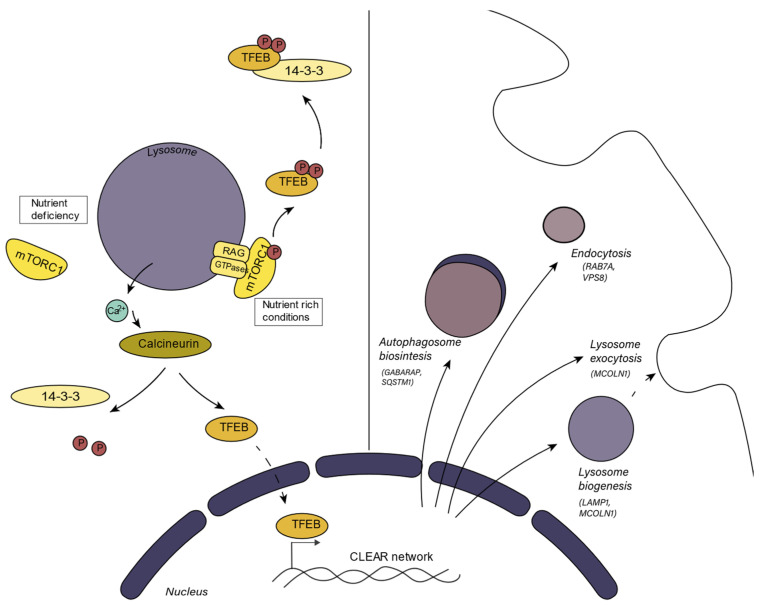
An overview of TFEB regulation through mTORC1 and calcineurin activity. Under nutrient rich conditions, mTORC1 phosphorylates TFEB, which is sequestered in the cytosol by 14-3-3 proteins. Under nutrient deficiency, mTORC1 is inactive, and calcineurin eliminates TFEB phosphates, allowing nuclear translocation, where TFEB induces the expression of genes related to autophagy, endocytosis, lysosome exocytosis, and lysosome biogenesis.

**Figure 2 cells-11-01977-f002:**
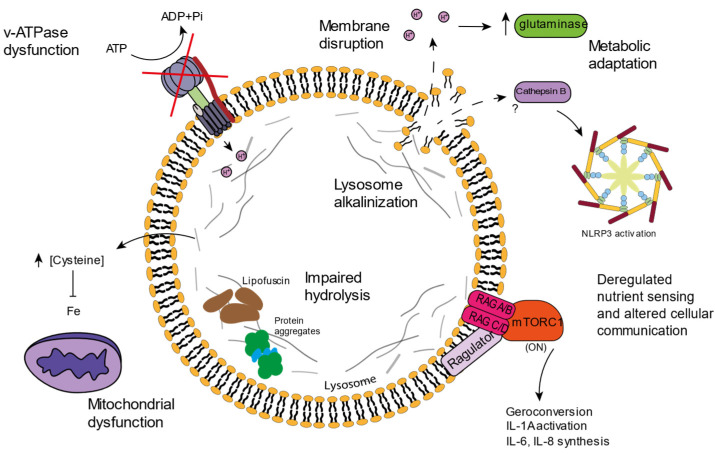
An overview of the lysosomal dysfunctions associated with aging or senescence. v-ATPase dysfunctions and membrane disruption impair the lysosomal acidification process, which shows increased alkalinization. Protons escaping the lysosome provoke cytosolic acidification, which alters cellular metabolism, and the presence of cathepsin B has been linked to NLRP3 inflammasome activation. Impaired hydrolysis inside the lysosome results in protein aggregation and lipofuscin storage. Incorrect amino acid storage is linked to mitochondrial dysfunction, and mTORC1 activation in the lysosome surface correlates with the synthesis of SASP molecules and geroconversion.

**Figure 3 cells-11-01977-f003:**
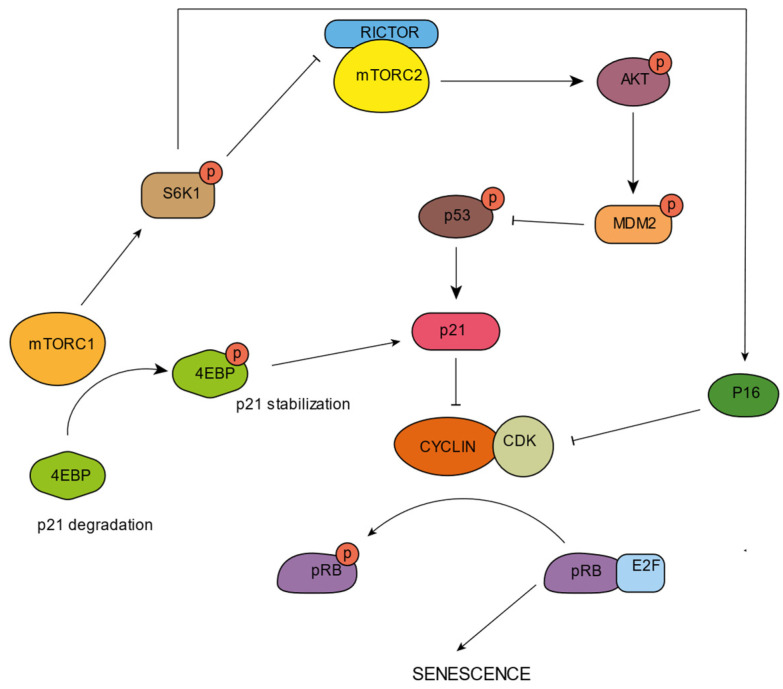
An overview of mTORC1 and mTORC2 on senescence control by the regulation of the p53/p21 and p16/pRb pathways. p21 is stabilized through mTORC1 dependent 4EBP phosphorylation, and through mTORC1 via S6K1 dependent mTORC2 inhibition, which has repercussions on p53 stabilization. p16 is activated downstream of mTORC1 by S6K1 phosphorylation, inhibiting the CDK–Cyclin complexes and impeding pRb phosphorylation. This results in the sequestration of E2F and the prevention of cell cycle progression, eventually leading to senescence.

## Data Availability

Not applicable.

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
