# Peer review of "Age-Related Lysosomal Dysfunctions"

_cells, 2022, doi:10.3390/cells11121977_

Round 1

Reviewer 1 Report

The topic of the review is interesting and relevant to the field. The authors contextualize well the lysosome and senescence background, although some information could still be added (e.g., CLEAR elements, other transcription factors besides TFEB, mTORC2). Since a major connection between the lysosome-autophagic machinery and senescence is via the mTORC1 (and mTORC2) regulation of p53/p21 and p16 a summary scheme of these molecular pathways could be presented. The authors should make an added effort to improve the flow of the major sections, in order to avoid it reading like an enumeration of observations. Particularly section 4 could be separated in several sub-sections, as there is a lot of different disperse information. The quality of the writing is high, so only minor English corrections are needed. Overall, the authors should try to improve the “thesis” behind the review to give more cohesion and better flow to the text. Please refer to some to some excellent reviews in the field of senescence and autophagy in cardiovascular diseases (e.g. 10.1093/cvr/cvy007) as examples of how to connect the field of lysosome/autophagy and age-related disorders.

Issues to be addressed:

  • Line 55: It could be useful to have a scheme depicting how mTORC1 and mTORC2 regulate p53, p21 and p16. See example in10.1007/s10522-020-09876-w
  • Line 77-81: avoid the repeated use of the term organelles.
  • Line 82: replace “where” by “in which”.
  • Line 145: reference missing after “…acidification process.”
  • Line 151: The CLEAR gene network should be mentioned in the main text and not only in the figure legend. It could also be mentioned that TFEB is not the only transcription factor regulating the CLEAR gene network (i.e. TFE3, MITF, TFEC etc). It is in fact a family of transcription factors MiT/TFE.
  • Line 165: reference missing after “…TFEB.” References should not be placed only at the end of the paragraph.
  • Line 175: the use of “narrowly” gives the impression that the processes are not very well connected, while I think you mean the opposite.
  • Line 188: you could also consider making a brief mention to the regulating role of mTORC2 somewhere in this section.
  • Line 303: “mTORC1” and not “TORC1”.
  • Line 330: reference missing after “autophagosome”.
  • Line 332: This section needs sub-sections as there is a lot of different information listed.
  • Line 367: the mechanism of NLRP3 inflammasome activation should be at least briefly described in the main text.
  • Line 389: crucial information missing: leakage of lysosomal proteins is thought to provide the second signal necessary for NLRP3 activation.
  • Line 443: It is crucial here to mention that this study was done in the yeast, otherwise this concept of lysosome-like vacuole falls out of the blue.
  • Line 446 and 450: these two paragraphs are examples of how sometimes the text can be quite fragmented. The authors should avoid simply listing observations. A contextualization linking these observations should be strengthened.
  • Line 470: which mechanisms of cell death? This is quite an important fact related to the role of lysosome dysfunction in ageing and deserves a bit further elaboration.
  • Line 512: C. elegans should be in italics.
  • Line 523: TFEB activation in excess may also lead to malignancies, which would be important to mention in order to properly contextualize these proposed therapies.

Reviewer 2 Report

This manuscript is an very well written review.

Comments: I suggest to include a discussión obout the role of lysosomal
dysfunction and iron accumulation during senescence.

Total iron content increased  during cellular senescence, reaching approximately 10-fold higher levels than young cells. As lysosomes participate in iron metabolism, it would be interesting to discuss about the role of lysosome dysfunction in iron accumulation and the consequences of this accumulation in the senescence process.

Round 2

Reviewer 1 Report

The authors have replied to all the points raised in a satisfactory manner. The manuscript is of sufficient quality to be published in the present form. Only minor language corrections may be necessary.